# Unhelmeted Riding, Drunk Riding, and Unlicensed Riding among Motorcyclists: A Population Study in Taiwan during 2011–2016

**DOI:** 10.3390/ijerph20021412

**Published:** 2023-01-12

**Authors:** Yen-Hsiu Liu, Bayu Satria Wiratama, Chung-Jen Chao, Ming-Heng Wang, Rui-Sheng Chen, Wafaa Saleh, Chih-Wei Pai

**Affiliations:** 1Graduate Institute of Injury Prevention and Control, College of Public Health, Taipei Medical University, Taipei 110, Taiwan; 2Department of Biostatistics, Epidemiology, and Population Health, Faculty of Medicine, Public Health and Nursing, Universitas Gadjah Mada, Yogyakarta City 55281, Indonesia; 3Department of Traffic Science, Central Police University, Taoyuan 333, Taiwan; 4Department of Traffic Management, Taiwan Police College, Taipei 116, Taiwan; 52nd District Headquarters, Yongji Station, Fire Department of Taipei City, Taipei 110, Taiwan; 6Transport Research Institute, Edinburgh Napier University, Edinburgh EH11 4DY, UK

**Keywords:** unhelmeted riding, drunk riding, unlicensed riding, ROR crashes

## Abstract

This study aimed to investigate the association between drunk riding, unhelmeted riding, unlicensed riding, and running-off-road (ROR) crashes. Multiple logistic regression was used to calculate the adjusted odds ratio (AOR) by using the National Taiwan Traffic Crash Dataset for 2011–2016. The results revealed that unhelmeted riding was associated with 138% (AOR = 2.38; CI (confidence interval) = 2.34–2.42) and 47% (AOR = 1.47; CI = 1.45–1.49) higher risks of drunk riding and unlicensed riding, respectively. The risk of unhelmeted riding increased with blood alcohol concentrations (BACs), and riders with the minimum BAC (0.031–0.05%) had nearly five times (AOR = 4.99; CI = 4.74–5.26) higher odds of unlicensed riding compared with those of riders with a negative BAC. Unhelmeted riding, drunk riding, and unlicensed riding were associated with 1.21 times (AOR = 1.21; CI = 1.13–1.30), 2.38 times (AOR = 2.38; CI = 2.20–2.57), and 1.13 times (AOR = 1.13; CI = 1.06–1.21) higher odds of ROR crashes, respectively. The three risky riding behaviours (i.e., unhelmeted riding, drunk riding, and unlicensed riding) were significantly related to ROR crashes. The risk of unhelmeted riding and ROR crashes increased with BACs.

## 1. Introduction

Motorcyclists comprise 28% of all global road traffic fatalities [1]. At a regional level, South East Asia has the highest proportion of motorcycle fatalities at 43% due to its considerably large motorcyclist population. A total of 14,342,361 motorcycles were registered in 2022 in Taiwan [2], suggesting that approximately 50% of the population owns a motorcycle. Official statistics in 2022 revealed that as high as 78% of all traffic fatalities were related to motorcyclists [2].

Research on injured body regions caused by helmet non-use has primarily focused on the head and cervical spine [3,4,5,6,7]. Wiznia et al. [8] concluded that unhelmeted riders were twice as likely to suffer head injuries and were almost twice as likely to die from their injuries. The protective effect of helmets has been well documented in the literature [9,10,11,12]. In particular, it was found to significantly reduce the relative risks of head injuries among motorcyclists [13,14,15,16,17,18].

Numerous studies have demonstrated that riding without a helmet was correlated with certain traffic violations among motorcyclists, such as red-light running [19,20]. Risk compensation hypothesis, nonetheless, has been used to explain how reduced risk perception (i.e., feeling safer) from helmet use may contribute to engagement in risky behaviours, such as speeding. Although some researchers have suggested that this effect appeared to be strong solely among cyclists who routinely wear helmets [21,22], a majority of researchers opposed this hypothesis [23,24]. Ouellet [25] investigated how risk compensation influenced helmeted motorcyclists’ behaviours by reviewing data from two studies conducted in Thailand and Los Angeles. Likewise, his findings were inconsistent with the hypothesis that the increased safety provided by motorcycle helmet use was offset by risky riding behaviours. In later studies, Bambach et al. [26] and Orsi et al. [27] noted that unhelmeted cyclists were more likely to engage in risky cycling behaviours, such as failing to comply with traffic-control signals and drunk cycling with blood alcohol concentrations (BACs) higher than 0.05%.

The literature also suggested that numerous risky riding behaviours were prevalent among intoxicated motorcyclists. For instance, intoxicated motorcyclists were less likely to wear a helmet compared with sober ones [28,29,30,31]. Furthermore, motorcyclists who rode under the influence of alcohol were more likely to call or text [32,33] and to engage in speeding and unlicensed riding [34,35]. A Thai study conducted by Kasantikul et al. [36] pointed out that drunk riders were more likely to be inattentive, run off the road, and violate traffic-control signals.

When reviewed together, these studies support that alcohol use and riding without a helmet were risk factors for severe and fatal injuries. Riders have also been observed to engage in several risk-taking behaviours, such as unlicensed riding [37,38]. However, such research has focused on the individual effect of alcohol use rather than that of riding without a helmet or riding without a license. Furthermore, running-off-road (ROR) crashes, which typically result in devastating injury outcomes [39,40], have also rarely been evaluated. ROR crashes occur when the motorcyclist leaves the travel lane and overturns or strikes natural or artificial objects, such as trees, bridge walls, poles, guardrails, embankments, and parked vehicles. These roadside barriers pose a significant risk to motorcycle riders, resulting in serious lower extremity and spinal injuries, and head injuries [40]. In Taiwan, ROR crashes had a high fatality percentage for drivers (26.4%) and an even higher percentage for motorcyclists (41.4%) [39]. The current research aims to fill in this research gap by investigating the relationship between unhelmeted riding, drunk riding, unlicensed riding, and ROR crashes.

## 2. Materials and Methods

### 2.1. Data Source

This study used crash data extracted from the National Taiwan Traffic Crash Dataset for the period from 2011 to 2016. The crash dataset, which is based on traffic-crash investigation reports completed by police crash investigators, includes two outcomes for injuries: number of deaths within 24 h and number of injuries. The dataset comprises two parts: crash information and crash victims. Crash information includes the crash location, time of crash, lighting, weather, road conditions, road type, traffic signal status, traffic lane, and crash type. Crash victim data include victim characteristics (such as age and sex), injury severity, primary anatomic injuries, BAC, vehicle type, helmet use, and license status.

Figure 1 presents the sample selection from the National Taiwan Traffic Crash Dataset. A total of 3,616,997 crashes over the period 2011–2016 were included in analyses. After exclusion of non-motorcycle crashes (*n* = 1,619,369) and cases with missing data (*n* = 25,883), 1,971,745 crashes involving motorcycles remained. Cases with missing data on age (*n* = 8987) were removed, resulting in 1,962,758 valid cases of motorcycle crashes. Among 1,962,758 motorcycle crashes, crashes involving unhelmeted riding accounted for 213,453 cases, and crashes involving helmeted riding accounted for 1,749,305 cases. Alcohol-involved crashes and non-alcohol-involved crashes comprised 111,825 and 1,850,933 cases, respectively. Drivers aged <18 years are classified as illegal drivers because of their ineligibility for a valid driver’s license in Taiwan; therefore, we removed data on those aged <18 years (*n* = 38,947). This resulted in a total of 1,923,811 motorcycle crashes, in which 1,629,025 crashes involved licensed riders and 284,786 crashes involved unlicensed riders. We then removed multi-vehicle motorcycle crashes (*n* = 1,812,875) from 1,962,758 motorcycle crashes, resulting in a dataset of 149,883 single-motorcycle crashes. Of these single-motorcycle crashes, 6527 were ROR crashes, and 143,356 were other crashes.

### 2.2. Definitions of Variables

The dependent variables in this study were helmet use (yes: helmeted, or no: unhelmeted), BAC (negative: <0.03%, or positive: 0.031–0.05%, 0.051–0.16%, >0.16%), license status (licensed: with a valid driver license, or unlicensed: without a valid driver license), and crash type (running-off-road or other crashes). Variables collected in this study were divided into two parts: rider characteristics and environmental factors. Rider characteristics included sex (male and female), age (<18, 18–40, 41–64, and ≥65 years), and engine size (heavy motorcycles: >250 cc, or mopeds: ≤250 cc). Variables regarding crash environments were defined as follows: weather (clear: sunny day, or unclear: storms, strong winds, sandstorms, fog or smoke, snow, rain, and clouds), streetlight conditions (daylight, dark with lights, and dark), crash month (spring/summer: March–August, or autumn/winter: September–February), crash day (weekday or weekend), and crash time (midnight/early morning: 0000–0600 h, daytime peak: 0700–0800 and 1700–1800 h, daytime off-peak: 0900–1600 h, and night time: 1900–2300 h). Speed limit (urban: <50 km/h, or rural: ≥50 km/h) was defined based on the widely adopted classification system on urban and rural roadways in Taiwan [41].

### 2.3. Statistical Analysis

The distribution of dependent variables, namely risky riding behaviours (e.g., unhelmeted riding, drunk riding, and unlicensed riding) and ROR crashes, was compared using a set of variables (e.g., rider’s characteristics and environmental factors). Subgroup analysis was conducted for each risky riding behaviour and for ROR crashes. Chi-square tests for independence were used to examine the association between the independent variables and dependent variables. To minimise type II errors in variable selection and biased inferences, we adopted a cut-off *p* value of <0.2, which is the conventional level adopted in the literature [18,42,43] for the inclusion of variables in the multivariate logistic regression models. We used multiple logistic regression analysis with backward selection to calculate the adjusted odds ratio (AOR). The confidence level of 95% and alpha (α) of 0.05 were adopted. Multicollinearity among variables was assessed using Cramer’s V and chi-square independent tests.

## 3. Results

### 3.1. General Results

Table 1 illustrates the distribution of unhelmeted riding and drunk riding across the independent variables. During the period of 2011–2016 there were 1,962,758 motorcycle crashes, of which 213,453 (10.88%) were individuals riding without wearing a helmet and 111,825 (5.7%) were riding after drinking alcohol. More motorcycle crashes involved male motorcyclists (59.12%), riders aged 18–40 years (63.88%), heavy motorcycle riders (91.76%), and occurred on rural roadways with speed limits ≥50 km/h (72.95%), on weekdays (76.41%), and during daylight conditions (71.71%).

As many as 11.57% and 7.12% of male motorcyclists engaged in unhelmeted riding and drunk riding, respectively, which was higher than that of their female counterparts. The results were statistically significant by using the chi-square test (*p* < 0.001). Although merely 1.98% of riders aged <18 years were involved in motorcycle crashes, 17.12% of them were observed to ride unhelmeted. More than 25% of riders with a BAC of ≥0.16% engaged in unhelmeted riding, which was higher than that of those of any other BAC (*p* < 0.001). Approximately 12% of unhelmeted riders drank alcohol before riding, compared with 4.9% of helmeted riders. Unlicensed riding appeared to be a contributory factor to risky riding; 14.89% and 10.85% of unlicensed riders engaged in unhelmeted riding and drunk riding, respectively. Dark conditions without streetlights were overrepresented in unhelmeted riding and drunk riding crashes (15.44% and 16.66%, respectively). Approximately 14% and 17% of those travelling in midnight hours were riding unhelmeted and drunk, which was the highest proportion of crashes for all time periods.

As presented in Table 2, among 1,923,811 motorcycle crashes, 15.32% (*n* = 294,786) were riding without a license. As many as 16.72% of female motorcyclists were reported to ride without a license, which was higher than that of their male counterparts (14.34%) (*p* < 0.001). Although riders aged 18–40 years accounted for 65.18% of all motorcycle crashes, those aged ≥65 years (27.48%) had the highest tendency to ride without a valid license, followed by the age group of 41–64 years (17.34%) and 18–40 years (12.98%). The proportion of unlicensed riding cases was the highest for the nearly 50% of riders with a BAC of 0.031–0.05%. As many as 20.14% of unhelmeted riders exhibited unlicensed riding behaviours, which was higher than those indicated otherwise (*p* < 0.001). A disproportionately high percentage of those riding in the midnight hours (17.81%) and on urban roadways (16.8%) were riding without a license.

As presented in Table 3, among 149,883 single-motorcycle crashes, ROR crashes accounted for 6527 cases (4.35%) during the period of 2011–2016. A higher proportion of single-motorcycle crashes involved riders aged 18–40 years (63.47%). More single-motorcycle crashes occurred on weekdays (74.48%), in clear weather (76.63%), on rural roadways (66.69%), and in daylight conditions (63.08%). Although adolescent and elderly riders contributed the least to single-motorcycle crashes (1.91% and 8.17%, respectively), they were equally subject to ROR crashes (7.62% for adolescent riders and 7.35% for elderly riders). The risk of ROR crashes increased with an increase in BACs: 7.56% for 0.031–0.05%, 7.73% for 0.051–0.16%, and 9.2% for ≥0.16%. Moreover, 6.19% of unhelmeted riders and 5.62% of unlicensed riders were involved in ROR crashes, which was higher than that of helmeted riders (4.07%) and licensed riders (4.08%) (*p* < 0.001). As many as 9.75% of ROR crashes occurred in the dark without streetlights, followed by daylight conditions (4.45%) and dark conditions with streetlights (3.51%). Midnight hours (5.49%) and urban roadways (5.31%) were overrepresented in ROR crashes.

### 3.2. Model Estimation Results

Multivariate logistic regression models were used to estimate the odds of unhelmeted riding, drunk riding, unlicensed riding, and ROR crashes among motorcyclists after other variables had been controlled for.

As reported in Table 4, male riders were more likely to engage in unhelmeted riding (AOR = 1.15; CI (confidence interval) = 1.14–1.16), and drunk riding (AOR = 2.21; CI = 2.18–2.24) than female riders. By contrast, the risk of unlicensed riding increased by 26% (AOR = 1.26; CI = 1.25–1.27) among riders who were female. The age group of ≥65 years was associated with 1.44 times (AOR = 1.44; CI = 1.42–1.46) and 2.53 times (AOR = 2.53; CI = 2.50–2.57) higher odds of unhelmeted riding and unlicensed riding, respectively. Riders aged 41–64 years had 2.33 times (AOR = 2.33; CI = 2.30–2.36) higher odds of drunk riding. Higher probabilities of ROR crashes were observed in both riders aged <18 years (AOR = 1.87; CI = 1.60–2.18) and ≥65 years (AOR = 1.89; CI = 1.74–2.04). In comparison with riding heavy motorcycles, riding mopeds was associated with 36% (AOR = 1.36; CI = 1.33–1.38) higher odds of drunk riding and 10% (AOR = 1.10; CI = 1.09–1.12) higher odds of unlicensed riding.

Riders with a BAC of ≥0.16% had 152% (AOR = 2.52; CI = 2.45–2.60) and 138% (AOR = 2.38; CI = 2.20–2.57) higher probability of engaging in unhelmeted riding and being involved in ROR crashes, respectively. By contrast, riders with the minimum alcohol consumption (BAC of 0.031%–0.05%) had nearly five times (AOR = 4.99; CI = 4.74–5.26) higher odds of unlicensed riding compared with those with a negative BAC. Unhelmeted riding was associated with a 138% (AOR = 2.38; CI = 2.34–2.42) higher risk of drunk riding and a 47% (AOR = 1.47; CI = 1.45–1.49) higher risk of unlicensed riding. Unhelmeted riding was associated with a 21% (AOR = 1.21; CI = 1.13–1.30) increase in the risk of ROR crashes. Unlicensed riders were 1.47 times (AOR = 1.47; CI = 1.45–1.49) and 2.61 times (AOR = 2.61; CI = 2.57–2.65) more likely to engage in unhelmeted riding and drunk riding than licensed riders. Unlicensed riders were estimated to have increased odds of being involved in ROR crashes of 13% (AOR = 1.13; CI = 1.06–1.21).

Environmental factors, such as streetlight condition and crash time, were associated with risky riding behaviours and ROR crashes. Riding in the dark without streetlights was correlated with 39% (AOR = 1.39; CI = 1.34–1.44) and 220% (AOR = 3.20; CI = 3.08–3.33) increased probabilities of unhelmeted riding and drunk riding, respectively. A 164% (AOR = 2.64; CI = 2.39–2.92) higher risk of ROR crashes for motorcyclists riding in the dark without streetlights was noted. Riding during midnight hours was associated with 1.24 times (AOR = 1.24; CI = 1.21–1.26), 3.64 times (AOR = 3.64; CI = 3.57–3.72), and 1.54 times (AOR = 1.54; CI = 1.36–1.63) more likely to engage in unhelmeted riding, drunk riding, and unlicensed riding, respectively. Similarly, riders travelling during midnight hours were more likely to have ROR crashes (AOR = 1.36; CI = 1.24–1.50) compared with those travelling in the daytime.

## 4. Discussion

One of the primary results of this study is that unhelmeted motorcyclists were significantly more likely to engage in drunk riding and unlicensed riding. Similar findings have been reported in cyclist safety literature [26,27,44] that unhelmeted cyclists were more likely to engage in risky riding behaviours. Our finding is consistent with relatively few studies focusing on unhelmeted motorcyclists, which suggested that unhelmeted riders were inclined to have BACs over the legal limit [45,46]. A study by Safiri et al. [38] confirmed that a greater percentage of people who wore helmets possessed a license, compared with those not wearing helmets. A possible explanation for our finding is that failing to wear a helmet might reflect a pattern of poor judgment or risky behaviours [45]. Furthermore, wearing a helmet served as a rule-compliance indicator [47]; therefore, unlike helmeted riders, unhelmeted riders were less likely to agree with mandatory traffic policy.

Consistent with previous research [30,48,49], risk compensation is not evident in our study. One argument proposed in opposition to the mandatory helmet law is based on the concept of risk compensation, implying that individuals equipped with protective gear, such as helmets, tend to act in a riskier manner due to the perception of increased protection provided by the helmet [50]. Contrary to this hypothesis, unhelmeted riders in our study appeared to exhibit more risk-taking behaviours by riding under the influence of alcohol and without a valid license. Countermeasures, such as stricter law enforcement, and more stringent punishments, such as fines and license suspension for unhelmeted riders, should be considered [51]. Future efforts should also consider designing intensive educational campaigns for unhelmeted riders which focus on the proper use of helmets [52].

Our finding also revealed that risky riding behaviours were common among motorcyclists who have engaged in at least one other form of risky riding. For example, those riding without a license were more prone to engage in drunk riding and unhelmeted riding, which is consistent with several previous studies [53,54,55,56,57,58,59,60]. Similarly, coupled with previous research [28,29,30,31,35,37,61,62,63,64,65,66], we found that riders engaging in drunk riding were more likely to ride unhelmeted and without a license.

Interestingly, the probability of risky riding behaviours varies depending on BACs. The risk of unhelmeted riding increased with increases in BAC; BAC ≥ 0.16% resulted in the highest odds of unhelmeted riding. However, those riding with a low BAC of 0.031–0.05% rather than those with a high BAC had the highest odds of unlicensed riding. This finding raises another issue that a low BAC can in fact pose a safety threat to road users by engaging in risky riding behaviours [67,68]. Although studies have proposed lowering the legal BAC limit imposed on motorcycle riders to reduce alcohol-related crash risks [67], zero-tolerance regulation was recommended for the entire population [69]. To reduce risky riding behaviours, such as drunk riding or unlicensed riding, improved detection methods and penalty increases may constitute an effective countermeasure [64].

Our study identified three risky riding behaviours as risk factors for ROR crashes, namely unhelmeted riding, drunk riding, and unlicensed riding. This finding is consistent with previous research [36,64,70,71,72]. In particular, we observed a positive association between BACs and ROR crashes, suggesting that the odds of ROR crashes elevated as BACs increased and reached the maximum likelihood at ≥0.16%. In addition to existing engineering measures, such as crash barriers [73,74], that have been adopted to prevent ROR crashes for all motorised vehicles, stricter enforcement to prosecute unhelmeted or drunk riding may reduce motorcycle crashes in general and ROR crashes in particular.

Elderly riders aged over 65 years were significantly more likely to engage in unhelmeted riding, drunk riding, and unlicensed riding. Previous research has attributed the high crash involvement of elderly motorcyclists to age-related declines in cognitive functions and poor reaction time [75,76]. Furthermore, injuries sustained by the elderly tended to be severe because of their deteriorating health condition [77,78]. In light of our findings, it is evident that injuries and risky riding behaviours among elderly motorcyclists should not be overlooked in this ageing generation. One intervention that has targeted elderly motorcyclists (and elderly vehicle drivers generally) is the license renewal law—riders aged 75 or older are mandated to renew their license every 3 years in Taiwan, dependant on whether they pass the physical (e.g., visual acuity) and cognitive examinations (e.g., spatial cognition). Such a law, which aims to reduce private vehicle dependence among the elderly, should be enforced more extensively.

In line with previous study [73] that has identified elderly riders (≥65 years) as a risk factor for ROR crashes, our findings revealed that riders aged <18 years exhibited a similar risk. Given the high risk of being killed among elderly motorcyclists involved in ROR crashes [78,79], our finding highlights the importance of establishing an effective intervention targeting both elderly and adolescent riders.

Our findings suggested that midnight hours were associated with unhelmeted riding, drunk riding, unlicensed riding, and ROR crashes. Our results regarding unhelmeted riding and drunk riding were consistent with previous findings [66,80,81,82]. To attenuate these risky behaviours, tightened law enforcement and the use of alcohol checkpoints during midnight hours should be considered. Our finding that midnight hours were a risk factor for ROR crashes is consistent with Montella et al. [39], who identified poor visibility at midnight as a contributory factor to ROR crashes. This underscores the importance of improving visibility during midnight hours at roadways where ROR crashes are likely to occur.

This research has several limitations. First, data on ROR crashes may be susceptible to underreporting. The police-reported crash dataset used in our study provided reliable data on unhelmeted riding, drunk riding, and unlicensed riding due to the mandatory BAC test imposed on motorcyclists and routinely collected information by the police in Taiwan, such as helmet use and license status. However, previous research [83,84] has noted the substantial underreported motorcycle crashes in nonfatal casualties and single-vehicle crashes, which we used to extract data on ROR crashes. Second, certain variables were unavailable in the National Taiwan Traffic Crash Dataset, such as injury severity, injured body regions, and trauma caused by risky riding. Therefore, future research may attempt to link multiple datasets, for instance, linking police records (e.g., the National Taiwan Traffic Crash Dataset) with hospital records (e.g., the National Health Insurance Research Database) to generate more comprehensive data on ROR crashes as well as injury-related information. Third, it is beyond the scope of the present study to examine whether motorcyclists were concurrently engaging in these risky behaviours. Future research may attempt to investigate the joint effects of unhelmeted riding, drunk riding, and unlicensed riding on ROR crashes. Finally, our findings may only be applied to Taiwan. The generalisability of our findings to different contexts should be exercised with caution until other studies using data from other jurisdictions have been reported.

## 5. Conclusions

This study demonstrated that unhelmeted riders were significantly more likely to engage in drunk riding and unlicensed riding. Additionally, risky riding behaviours were common among motorcyclists who have engaged in at least one other form of risky riding. In particular, we observed that the risk of unhelmeted riding increased with increases in BAC, and riders with the minimum BAC of 0.031%–0.05% were five times more likely to engage in unlicensed riding compared with sober riders. Moreover, unhelmeted riding, drunk riding, and unlicensed riding were found to be significantly related to ROR crashes. Notably, a positive association between BACs and ROR crashes was evident. We also identified that elderly motorcyclists and riding during midnight hours/early mornings were significantly associated with higher odds of engaging in risky riding and being involved in ROR crashes. Countermeasures should therefore target the older age group and focus on augmentation of law enforcement at midnight.

## Figures and Tables

**Figure 1 ijerph-20-01412-f001:**
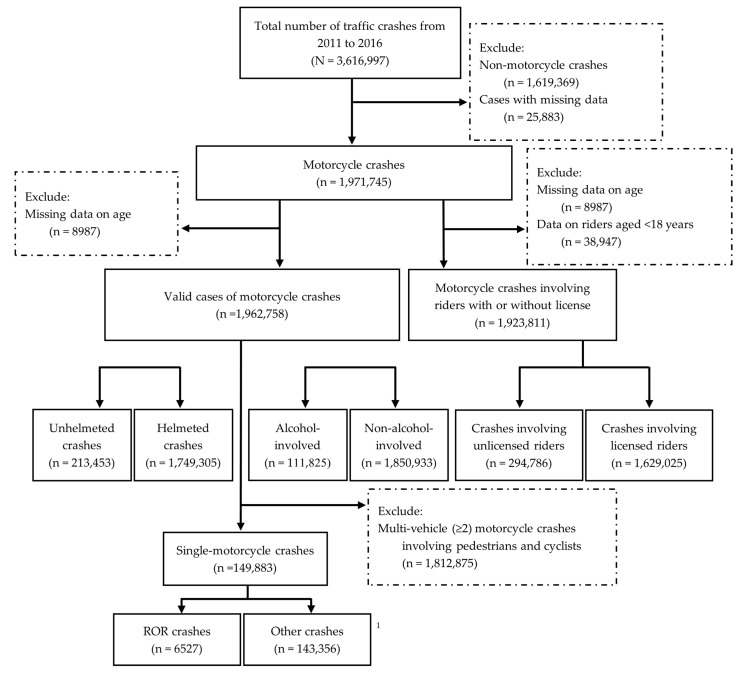
Sample selection process. ^1^ Other single-motorcycle crashes involve: spillover crashes, hit other objects crashes, and hit traffic facilities crashes.

**Table 1 ijerph-20-01412-t001:** Distribution of unhelmeted riding and drunk riding.

Variable	*n*	Helmeted Riding	Unhelmeted Riding	χ^2^ Test *p* Value	Alcohol-Involved	Non-Alcohol-Involved	χ^2^ Test *p* Value
		*n* (%)	*n* (%)		*n* (%)	*n* (%)	
Total	1,962,758	1,749,305 (89.12)	213,453 (10.88)		111,825 (5.70)	1,850,933 (94.30)	
Rider’s sex							
Male	1,160,436 (59.12)	1,026,160 (88.43)	134,276 (11.57)	<0.001	82,640 (7.12)	1,077,796 (92.88)	<0.001
Female	802,322 (40.88)	723,145 (90.13)	79,177 (9.87)		29,185 (3.64)	773,137 (96.36)	
Rider’s age							
<18	38,947 (1.98)	32,281 (82.88)	6666 (17.12)	<0.001	2118 (5.44)	36,829 (94.56)	<0.001
18–40	1,253,898 (63.88)	1,128,913 (90.03)	124,985 (9.97)		54,642 (4.36)	1,199,256 (95.64)	
41–64	512,677 (26.12)	453,235 (88.41)	59,442 (11.59)		44,496 (8.68)	468,181 (91.32)	
≥65	157,236 (8.01)	134,876 (85.78)	22,360 (14.22)		10,569 (6.72)	146,667 (93.28)	
Blood alcohol concentration (BAC)							
No	1,901,022 (96.85)	1,700,959 (89.48)	200,063 (10.52)	<0.001	-	-	-
0.031–0.05%	6286 (0.32)	5414 (86.13)	872 (13.87)				
0.051–0.16%	28,997 (1.48)	23,285 (80.30)	5712 (19.70)				
≥0.16%	26,453 (1.35)	19,647 (74.27)	6806 (25.73)				
Helmet use							
No	213,453 (10.88)	-	-	-	26,193 (12.27)	187,260 (87.73)	<0.001
Yes	1,749,305 (89.12)				85,632 (4.90)	1,663,673 (95.10)	
License status							
Unlicensed	333,655 (17.00)	283,981 (85.11)	49,674 (14.89)	<0.001	36,212 (10.85)	297,443 (89.15)	<0.001
Licensed	1,629,103 (83.00)	1,465,324 (89.95)	163,779 (10.05)		75,613 (4.64)	1,553,490 (95.36)	
Engine size							
Heavy motorcycle	1,801,047 (91.76)	1,606,217 (89.18)	194,830 (10.82)	<0.001	99,673 (5.53)	1,701,374 (94.47)	<0.001
Moped	161,711 (8.24)	143,088 (88.48)	18,623 (11.52)		12,152 (7.51)	149,559 (92.49)	
Speed limit							
Urban (<50 km/h)	530,859 (27.05)	476,906 (89.84)	53,953 (10.16)	<0.001	34,226 (6.45)	496,633 (93.55)	<0.001
Rural (≥50 km/h)	1,431,899 (72.95)	1,272,399 (88.86)	159,500 (11.14)		77,599 (5.42)	1,354,300 (94.58)	
Streetlight condition							
Dark	25,957 (1.32)	21,948 (84.56)	4009 (15.44)	<0.001	4325 (16.66)	21,632 (83.34)	<0.001
Dark with lights	529,343 (26.97)	467,414 (88.30)	61,929 (11.70)		45,177 (8.53)	484,166 (91.47)	
Daylight	1,407,458 (71.71)	1,259,943 (89.52)	147,515 (10.48)		62,323 (4.43)	1,345,135 (95.57)	
Crash time							
Daytime off-peak	851,346 (43.37)	760,672 (89.35)	90,674 (10.65)	<0.001	36,494 (4.29)	814,852 (95.71)	<0.001
Daytime peak	627,256 (31.96)	563,549 (89.84)	63,707 (10.16)		26,917 (4.29)	600,339 (95.71)	
Night-time	373,884 (19.05)	330,399 (88.37)	43,485 (11.63)		29,406 (7.87)	344,478 (92.13)	
Midnight	110,272 (5.62)	94,685 (85.86)	15,587 (14.14)		19,008 (17.24)	91,264 (82.76)	

**Table 2 ijerph-20-01412-t002:** Distribution of unlicensed riding.

Variable	*n*	Licensed Riding	Unlicensed Riding	χ^2^ Test *p* Value
		*n* (%)	*n* (%)	
Total	1,923,811	1,629,025 (84.68)	294,786 (15.32)	
Rider’s sex				
Male	1,129,551 (58.71)	967,584 (85.66)	161,967 (14.34)	<0.001
Female	794,260 (41.29)	661,441 (83.28)	132,819 (16.72)	
Rider’s age				
18–40 years	1,253,898 (65.18)	1,091,199 (87.02)	162,699 (12.98)	<0.001
41–64 years	512,677 (26.65)	423,792 (82.66)	88,885 (17.34)	
≥65 years	157,236 (8.17)	114,034 (72.52)	43,202 (27.48)	
BAC				
No	1,862,860 (96.83)	1,585,771 (85.13)	277,089 (14.87)	<0.001
0.031–0.05%	6121 (0.32)	3148 (51.43)	2973 (48.57)	
0.051–0.16%	28,536 (1.48)	19,803 (69.40)	8733 (30.60)	
≥0.16%	26,294 (1.37)	20,303 (77.22)	5991 (22.78)	
Helmet use				
No	206,787 (10.75)	163,771 (79.20)	43,016 (20.80)	<0.001
Yes	1,717,024 (89.25)	1,465,254 (85.34)	251,770 (14.66)	
Engine size				
Heavy motorcycle	1,763,729 (91.68)	1,501,180 (85.11)	262,549 (14.89)	<0.001
Moped	160,082 (8.32)	127,845 (79.86)	32,237 (20.14)	
Speed limit				
Urban (<50 km/h)	521,183 (27.09)	433,601 (83.20)	87,582 (16.80)	<0.001
Rural (≥50 km/h)	1,402,628 (72.91)	1,195,424 (85.23)	207,204 (14.77)	
Streetlight condition				
Dark	25,111 (1.31)	21,296 (84.81)	3815 (15.19)	<0.001
Dark with lights	513,894 (26.71)	442,174 (86.04)	71,720 (13.96)	
Daylight	1,384,806 (71.98)	1,165,555 (84.17)	219,251 (15.83)	
Crash time				
Daytime off-peak	837,428 (43.53)	704,941 (84.18)	132,487 (15.82)	<0.001
Daytime peak	617,137 (32.08)	526,220 (85.27)	90,917 (14.73)	
Night-time	362,380 (18.84)	310,028 (85.55)	52,352 (14.45)	
Midnight	106,866 (5.55)	87,836 (82.19)	19,030 (17.81)	

**Table 3 ijerph-20-01412-t003:** Distribution of run-off-road (ROR) crashes.

Variable	*n*	ROR	Others	χ^2^ Test *p* Value
		*n* (%)	*n* (%)	
Total	149,883	6527 (4.35%)	143,356 (95.65%)	
Rider’s sex				
Male	96,987 (64.71)	4588 (4.73%)	92,399 (95.27%)	0.004
Female	52,896 (35.29)	1939 (3.67%)	50,957 (96.33%)	
Rider’s age				
<18 years	2861 (1.91)	218 (7.62%)	2643 (92.38%)	<0.001
18–40 years	95,128 (63.47)	3437 (3.61%)	91,691 (96.39%)	
41–64 years	39,656 (26.46)	1973 (4.98%)	37,683 (95.02%)	
≥65 years	12,238 (8.17)	899 (7.35%)	11,339 (92.65%)	
BAC				
No	130,763 (87.24)	4878 (3.73%)	125,885 (96.27%)	<0.001
0.031–0.05%	622 (0.41)	47 (7.56%)	575 (92.44)	
0.051–0.16%	6976 (4.65)	539 (7.73%)	6437 (92.27)	
≥0.16%	11,522 (7.69)	1063 (9.20%)	10,459 (90.80)	
Helmet use				
No	19,931 (13.30)	1234 (6.19%)	18,697 (93.81%)	<0.001
Yes	129,952 (86.70)	5293 (4.07%)	124,659 (95.93%)	
License status				
Unlicensed	26,545 (17.71)	1491 (5.62%)	25,054 (94.38%)	<0.001
Licensed	123,338 (82.29)	5036 (4.08%)	118,302 (95.92%)	
Engine size				
Heavy motorcycle	137,931 (92.03)	5942 (4.31%)	131,989 (95.69%)	0.123
Moped	11,952 (7.97)	585 (4.89%)	11,367 (95.11%)	
Speed limit				
Urban (<50 km/h)	49,925 (33.31)	2650 (5.31%)	47,275 (94.69%)	<0.001
Rural (≥50 km/h)	99,958 (66.69)	3877 (3.88%)	96,081 (96.12%)	
Streetlight condition				
Dark	5948 (3.97)	580 (9.75%)	5368 (90.25%)	<0.001
Dark with lights	49,389 (32.95)	1736 (3.51%)	47,653 (96.49%)	
Daylight	94,546 (63.08)	4211 (4.45%)	90,335 (95.55%)	
Crash time				
Daytime off-peak	56,685 (37.82)	2761 (4.87%)	53,924 (95.13%)	<0.001
Daytime peak	40,709 (27.16)	1397 (3.43%)	39,312 (96.67%)	
Night-time	34,726 (23.17)	1394 (4.01%)	33,332 (95.99%)	
Midnight	17,763 (11.85)	975 (5.49%)	16,788 (94.51%)	

**Table 4 ijerph-20-01412-t004:** Multivariate logistic regression analyses.

Variable	Unhelmeted Riding		Drunk Riding		Unlicensed Riding		ROR Crashes	
	OR (95% CI)	*p* Value	OR (95% CI)	*p* Value	OR (95% CI)	*p* Value	OR (95% CI)	*p* Value
Rider’s sex								
Male	1.15 (1.14, 1.16)	<0.01	2.21 (2.18, 2.24)	<0.01	Ref	<0.01	1.09 (1.03, 1.15)	0.004
Female	Ref		Ref		1.26 (1.25, 1.27)		Ref	
Rider’s age								
<18	1.29 (1.26, 1.33)	<0.01	0.42 (0.40, 0.44)	<0.01	-	-	1.87 (1.60, 2.18)	<0.001
18–40	Ref		Ref		Ref		Ref	
41–64	1.16 (1.15, 1.17)	<0.01	2.33 (2.30, 2.36)	<0.01	1.30 (1.29, 1.31)	<0.01	1.16 (1.10, 1.23)	<0.001
≥65	1.44 (1.42, 1.46)	<0.01	1.45 (1.41, 1.48)	<0.01	2.53 (2.50, 2.57)	<0.01	1.89 (1.74, 2.04)	<0.001
Engine size								
Heavy motorcycle	Ref		Ref		Ref		Ref	
Moped	1.00 (0.98, 1.02)	0.864	1.36 (1.33, 1.38)	<0.01	1.10 (1.09, 1.12)	<0.01	0.93 (0.85, 1.02)	0.123
BAC								
No	Ref		-	-	Ref		Ref	
0.031–0.05%	1.13 (1.05, 1.21)	<0.01			4.99 (4.74, 5.26)	<0.01	1.89 (1.40, 2.55)	<0.001
0.051–0.16%	1.77 (1.72, 1.82)	<0.01			2.52 (2.46, 2.59)	<0.01	1.97 (1.79, 2.17)	<0.001
≥0.16%	2.52 (2.45, 2.60)	<0.01			1.70 (1.65, 1.75)	<0.01	2.38 (2.20, 2.57)	<0.001
Helmet use								
No	-	-	2.38 (2.34, 2.42)	<0.01	1.47 (1.45, 1.49)	<0.01	1.21 (1.13, 1.30)	<0.001
Yes			Ref		Ref		Ref	
License status								
Unlicensed	1.47 (1.45, 1.49)	<0.01	2.61 (2.57, 2.65)	<0.01	-	-	1.13 (1.06, 1.21)	<0.001
Licensed	Ref		Ref				Ref	
Speed limit								
Urban (<50 km/h)	Ref		1.19 (1.17, 1.21)	<0.01	1.15 (1.14, 1.16)	<0.01	1.26 (1.20, 1.33)	<0.001
Rural (≥50 km/h)	1.13 (1.12, 1.14)	<0.01	Ref		Ref		Ref	
Streetlight condition								
Dark	1.39 (1.34, 1.44)	<0.01	3.20 (3.08, 3.33)	<0.01	0.94 (0.91, 0.98)	<0.01	2.64 (2.39, 2.92)	<0.001
Dark with lights	1.09 (1.07, 1.11)	<0.01	1.73 (1.70, 1.77)	<0.01	1.09 (1.08, 1.11)	<0.01	1.36 (1.24, 1.49)	<0.001
Daylight	Ref		Ref		Ref		Ref	
Crash time								
Daytime off-peak	1.06 (1.05, 1.07)	<0.01	1.11 (1.095, 1.13)	<0.01	1.08 (1.05, 1.12)	<0.01	1.29 (1.20, 1.38)	<0.001
Daytime peak	Ref		Ref		Ref		Ref	
Night-time	1.05 (1.04, 1.07)	<0.01	1.32 (1.29, 1.35)	<0.01	1.15 (1.11, 1.19)	<0.01	1.15 (1.04, 1.27)	0.005
Midnight	1.24 (1.21, 1.26)	<0.01	3.64 (3.57, 3.72)	<0.01	1.54 (1.36, 1.63)	<0.01	1.36 (1.24, 1.50)	<0.001

## Data Availability

The data used for this study are available by request to the authors.

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
