# Peer review of "Unhelmeted Riding, Drunk Riding, and Unlicensed Riding among Motorcyclists: A Population Study in Taiwan during 2011–2016"

_ijerph, 2023, doi:10.3390/ijerph20021412_

Round 1

Reviewer 1 Report

ijerph-2057258

Unhelmeted Riding, Drunk Riding, and Unlicensed Riding Among Motorcyclists: A Population Study in Taiwan During 2011–2016

This paper aimed to investigate the association between drunk riding, unhelmeted riding, unlicensed riding, and running-off-road crashes. The authors had an access to the National Taiwan Traffic Crash Dataset for the period from 2011 to 2016 and were able to analyse a large set of data. Modelling through a multiple logistic regression is proposed.

This paper is very well-written and clearly organized that makes a smooth reading possible.

The references are in a large number so that the paper is well-documented.

I have some general comments and specific ones.

P2§49-62: please select only references with motorcycle helmet, or please specify the results are obtained for cyclists. Here remains a confusion to clarify for the reader.

P2§57-58: I do not see where is the inconsistency. At the opposite, the risk compensation theory could drive overcompensation that remains possible. It is not new, see Peltzman (1975) for the seminal work on the safety belt in the USA.

General statistics:

Please justify the fact you chose only three categories for age and why you chose these categories

Same question for the seasons (only two, why?)

Table 4 why the reference is changing for Speed Limit and Weather variables. Is it an error?

Concerning the different tables, I am note sure information about season, weather, and day are very useful. They are slightly discussed. I suggest to delete such information from the tables for focusing upon the essential.

Table 4 especially could be reorganized through categories of variables such context, individual characteristics, risk or behaviours.

Conclusion and abstract

Please avoid such a formulation “unhelmeted riding significantly increased the risk of drunk riding and unlicensed riding”. Indeed, it is assumed there is a kind of causality while you show it is a statistical association. Please rewrite with such word such as “engage” “associated with” you use previously through the paper. You have to harmonize and avoid a formulation tending towards causality.

My general comment concerns an issue on which the paper is not very clear.

It has to be distinguished risk taking from risk compensation. Risk taking is more appropriate for this paper. Risk compensation assumes the same individual exposed to a modification changes her behaviour. In the present case, it is not possible to test this hypothesis, so that the most appropriate solution is to avoid the evocation of that effect through the discussion, but the authors provides any additional information. Second there are different risk compensation mechanisms: micro one such the change of behaviour induced by the wearing of a helmet, and macro or general one, such the change of behaviour is induced by the combinations of different changes (See Wilde 1976?). Third, risk compensation does not mean a negative final effect: the wearing of helmet can induce a higher risk taking but not enough to fully compensate the safety effect of the helmet. Under, full and over compensation are possible while they belong all to the risk compensation mechanism. Fourth, risk compensation or risk change measures are a true challenge, but the present article can only state some associations. Consequently, I would suggest to keep the idea of an association of more or less high-risk variables with some other ones, that could be explained more by risk taking than using potential explanations which are not proved, that does not mean they do not exist.

It remains I highly appreciated the paper that deserves publication by taking into consideration these minor suggestions.

Reviewer 2 Report

This paper aims at investigating correlation between drunk riding, unhelmeted riding, unlicensed riding, and ROR collisions. In general, the topic does not look new but may be the first based on Taiwan data. The authors did a good job processing and analysing the data and drawing insights of it.

My biggest concern with this paper is the narrative is based on that a driver is not wearing a helmet will "cause" drunk riding. This is evident when it is said in the abstract "The results revealed that unhelmeted riding increased the risk of drunk riding and 23 unlicensed riding by 138%" This is a big issue in my opinion. There may be a correlation, but it does not mean that not wearing the helmet will CAUSE drunk riding. Again in the abstract, “Riding without a helmet may induce drunk riding and unlicensed riding” I totally disagree with this statement.

I think it should be said that unhelmeted drivers will likely be drunk and unlicensed. This should be changed within the entire manuscript as it may lead to misunderstanding.

Line 74 and 75, please show some stats from this previous research (ref 39 and 40) regarding the percentage of ROR of all crashes and how this ROR contribute to injuries and fatalities compared to other collision types.

Line 94 and 95, I disagree with having a collision type as unhelmeted riding unlike alchohol which can be a cause of a crash. Unhelmeted riding is a cause of death or severe injury not the crash itself. Unhelmeted riding is not a cause. Alcohol consumption is a cause of crash in general and may lead to sever injuries or fatal collisions. This idea should be kept in mind and adjusted throughout the manuscript and the narrative should be tweaked accordingly.

Figure 1 shows other crashes. Please provide examples of these crashes. Also, it is better to provide percentages beside the numbers for better illustrations.

Comment on chi square results in Tables 1, 2, and 3.

There is another issue in the discussion in the third point of the future research in lines 367 – 370. It is very confusing to me and led that I question what is the aim of this paper!

Reviewer 3 Report

Traffic accidents are still a current topic in road transport. The article is prepared at an appropriate scientific level. Appropriate statistical tools are used for data processing. To increase the level of a scientific article, it would be appropriate to: - use current data for today's year. It is not appropriate to publish data from 2011-2016 in 2022.

Reviewer 4 Report

The survey covers statistical data for several years. Their volume is completely sufficient for the statistical analysis of the interaction of various factors. The presentation of the results in tables allows you to clearly see the numerical evaluations, at the same time it should be noted that the graphs would have been very useful in the visualization of the research results.

Reviewer 5 Report

This paper statistically investigates three motorcyclist riding behaviors using data collected during 2011-2016 in Taiwan. The paper is well organized, and I have the following questions and comments.

(1)   Please use care when describing statistical relationships, which should differ from causality. For example, the sentence “Riding without a helmet may induce drunk riding and unlicensed riding” implies causality, but the paper does not prove it.

(2)   Please explain “CI”. I assume it should be confidence interval.

(3)   I don’t see any measures regarding Cramér’s phi in the paper, though the authors claimed the method was used. It may not be a significant issue as it can be equivalent to a Chi-squared independence test.

(4)   The results for “spring/summer” and “autumn/winter” data sets in terms of crash month can be questionable. The p-value is rather large. Please provide additional clarity on the design of the Chi-squared test.

(5)   Did the authors have individual observations with measured blood alcohol concentrations? Or the observations only associated with the three BAC categories. If only three categories were applied, it might not be appropriate to conclude there is a “linear” relationship between run-off road crash risk and BAC.

Round 2

Reviewer 2 Report

no additional comments